# Using the Isometric Mid-Thigh Pull to Predict Three-Repetition Maximum Squat Values in Female Athletes

**DOI:** 10.3390/sports12090230

**Published:** 2024-08-26

**Authors:** Keely Pasfield, Nick Ball, Dale Wilson Chapman

**Affiliations:** 1Research Institute for Sport and Exercise, University of Canberra, Canberra 2617, Australia; keeley.pasfield@gmail.com (K.P.); nick.ball@canberra.edu.au (N.B.); 2Physiology, Australian Institute of Sport, Canberra 2617, Australia; 3Curtin School of Allied Health, Curtin University, Perth 6102, Australia

**Keywords:** prediction equation, lower body, maximal strength, training load, back squat

## Abstract

Prescribing correct training loads in strength- and power-based sports is essential to eliciting performance improvements for athletes. Concurrently, testing strength for the prescription of training loads should be accurate and safe with minimal disruption or fatigue inducement to the athlete. The purpose of this study was to develop a prediction equation in female athletes for the three-repetition maximum (3RM) squat using the isometric mid-thigh pull and basic anthropometric assessments that could be practically applied to support training prescriptions. Female athletes (*n* = 34) were recruited from netball, volleyball, basketball, and soccer across a spectrum of competitive standards. Each athlete’s weight, standing height, seated height, arm span, and biacromial breadth were recorded, and then, on separate occasions separated by at least 48 h, each athlete completed a 3RM squat test and an isometric mid-thigh pull (IMTP) assessment. IMTP variables of peak force and time-dependent force at 50, 100, 150, 200, and 250 ms, as well as anthropometric measures, were used to develop a prediction equation. Squat strength was low-to-moderately correlated with peak force (r = 0.386); force at 100 ms (r = −0.128), 150 ms (r = −0.040), and 200 ms (r = −0.034); standing height (r = 0.294); and biacromial breadth (r = −0.410). Stepwise multiple regression significantly (*p* < 0.05) explained 26% of the 3RM squat strength variation using peak force and force at 100 ms, resulting in the following equation: Predicted 3RM squat (kg) = [6.102 + (Peak Force × 0.002) − (Force@100 ms × 0.001)]^2^. The reported equation’s predictive accuracy was tested using the same testing protocols following 6–8 weeks of training in a sub-cohort of athletes (*n* = 14). The predicted and actual recorded 3RM values were not significantly (*p* = 0.313) different, supporting the use of the IMTP as a test that contributes informative values for use in a predictive equation for training prescription and thus reducing the testing and fatigue-inducing impost on female athletes. However, the 95% CI (−4.18–12.09) indicated predicted values could differ in excess of 10 kg. This difference could lead to an excessive load prescription for an athlete’s training program, indicating caution should be taken if using the described method to predict 3RM squat values for programming purposes.

## 1. Introduction

Improvements in strength rely on the meticulous prescription and long-term planning of exercises and loads by strength and conditioning professionals [1]. Exercises such as the back squat, deadlift, and weightlifting lifts or derivatives are commonly used by strength and conditioning coaches in training programs due to their translation to performance [2]. Prescribing the correct load at which these exercises are performed in the athletic population is necessary to elicit strength adaptations without inducing excessive fatigue or overtraining. Assessments are commonplace in strength and conditioning to measure an athletes training status, improvements gained and to inform prescription of loads for the desired response. Prescribed loads are often programmed as a percentage of the maximal load that an individual can lift for a given number of repetitions. In relation to the back squat, the highest power is produced at loads of 30–70% of 1RM with strength developed at ≥75% 1RM [3]. A commonly used lower-body strength assessment is the three-repetition maximum (3RM) squat, with many coaches using this result to prescribe load for exercises in their resistance training programs. However, due to the required maximal effort, a back squat strength test can elicit muscle soreness which may impact an athlete’s continued training [4]. 3RM squat testing can be time consuming with multiple trials and significant rest periods between efforts are required to achieve an accurate measurement [5]. As athletes are constantly improving their performance, strength can change in a short period of time, requiring frequent testing to ensure the desired load prescription accuracy [6]. Considering these disadvantages, developing a robust and time-efficient prediction of a 3RM squat would be of practical importance to the strength and conditioning field, in particular to monitor the athlete’s performance, their response to training, and as a basis for exercise load prescription. Prediction equations in other athletic assessments have been successfully developed and used for these purposes, such as maximal oxygen uptake predictions [7].

Lower body strength is a key quality for athletic performance in team sports involving rapid change of direction, jumping, and sprinting movements, which is developed through resistance training [8]. With the increased participation and professionalism of female athletes in sport, an increased focus on research specific to the female population is needed to improve individual performance and thus competition quality. Due to anatomical, neuromuscular, and hormonal differences between males and females, women in general display significantly lower strength levels than men in the athletic population when no normalisation processes are applied [9,10]. Anatomically, men typically are taller and have a broader skeletal structure capable of supporting a greater volume of muscle mass, which contributes to force production [11], and it has been recommended when making comparisons between sexes that relative strength based on body weight can account for most differences [12]. The lower levels and fluctuations of circulating testosterone because of the menstrual cycle in females may explain why adaptations to strength training can be more prominent in men [13]. In addition to variances in absolute strength levels, Mata et al. [3] reported men and women differ in their neuromuscular qualities that influence their rates of contraction, with women displaying a higher contribution of velocity rather than force to power output, suggesting further improvements in females may result from resistance training focused on strength development.

The isometric mid-thigh pull (IMTP) is an assessment with a strong relationship to dynamic activities involving lower body strength and explosive power [14] and is advantageous due to its high test-retest reliability [15]. The IMTP position is designed to replicate the mid-portion (2nd pull) of the clean, where the most force is generated [16,17]. Due to its strong correlation with dynamic movement, the IMTP has been investigated to predict performance in male sprinting [18] and female weightlifting [17], but its application to predict load in traditional strength and conditioning exercises, on which it is based, has not been explored. In addition, it may be a more time-efficient method for testing strength with less chance to cause injury or fatigue than maximal dynamic movements under load due to the limited range of motion and muscle length change required [19]. Considering the translation of strength and conditioning exercises to performance in strength and power-based sports [2], it appears fortuitous to investigate methods of prescribing correct training loads with minimal disruption to athletes.

The purpose of this study was to develop a prediction equation for the 3RM squat specific to female athletes, based on force–time measures from the IMTP and anthropometric characteristics. We hypothesise that a robust predictive equation for the back squat 3RM will include metrics of force capacity and anthropometry. The real-world application of this investigation is to provide an alternative test with predictive accuracy, and thus improve the prescription of load for resistance training exercises whilst minimising impact on training and fatigue accumulation that might result from traditional direct strength assessments.

## 2. Materials and Methods

### 2.1. Experimental Approach to the Problem

The study design is composed of two distinct parts: Part A involved a single observational design for data collection utilising athletes from a cross-section of competitive experience. The results were modelled in a stepwise multiple regression procedure to create a prediction equation for the 3RM squat using force–time measurements from the isometric mid-thigh pull (IMTP) and anthropometric characteristics. Part B was a within-athlete (repeated measure) observational design with the investigator partially blinded to the outcome, assessing the reliability and accuracy of the equation developed in Part A by comparing the predicted and actual 3RM squat values following a 6–8-week period of sport-specific training, as programmed and periodised by the specific strength and conditioning coach. 

### 2.2. Subjects

Thirty-four female athletes were recruited from a variety of sports, including track and field (national competitor), soccer (1st league state level), netball, basketball, and volleyball (all national development squads) to participate in Part A (Table 1). As per the categorisation of McKay et al. [20], the recruited athletes’ training and performance calibre are tier 2 and 3. A sub-cohort from the original thirty-four athletes was recruited based on consent and availability (*n* = 14) to participate in the follow-up testing for Part B. All athletes were currently competing in their relevant sport and engaged in a regular supervised resistance training program involving two or more training sessions per week. The athletes and prescribing strength and conditioning coaches confirmed verbally with the lead author that regular 2–5RM squatting was included in the resistance training program for each athlete, with programming facilitated by and strength monitoring conducted via repetition maximum testing. As such, the authors determined that familiarisation with the 3RM squat was not required as athletes were already familiar with the test. Any athlete presenting with an existing chronic condition such as diabetes, heart conditions, etc., a symptomatic acute illness, a lower body injury within the previous three months, or an upper body injury that impaired maximal performance of any of the required movements were excluded. Each athlete was provided with a verbal explanation of the study expectations and was given the opportunity to seek clarification, following which they each completed an informed consent with a parent/guardian signature obtained for those under the age of 18 years. Ethical permission to conduct the study was obtained from the University of Canberra Ethics Committee. 

### 2.3. Procedures

#### 2.3.1. Part A and Part B

##### Part A

Data collection was completed within a maximum seven-day and a minimum three-day period to minimise any potential training effect or the impact of multiple match day fatigue. The IMTP test was conducted 24–96 h prior to the 3RM squat to minimise the impact of fatigue and residual soreness on the movement depending on the weekly training structure, with the anthropometric measures collected prior to either of these assessments. Athletes were instructed to refrain from consuming caffeine 4 h prior and vigorous activity 24 h prior to testing. Athletes were also instructed to ensure they were adequately rested, hydrated, and had consumed their last meal 30 min–4 h before engaging in strength testing. Athletes completed a standard warm-up protocol prior to either the IMTP or 3RM squat testing that included cycling at a comfortable pace (60–70RPM) for 60 s, followed by four 10 s sprint efforts every 50 s, and a dynamic warm-up comprising one set of each of walking lunges × 10 m, high knees × 10 m, butt kicks × 10 m, and body weight squats × 10.

##### Part B

All procedures from Part A were repeated following a 6–8-week period of training, during which it was assumed strength training adaptations were likely to have occurred [21]. However, to ensure that no investigator bias influenced the 3RM results, the second round of 3RM squat testing was completed with the primary researcher blinded to the results. These tests were conducted by suitably qualified external associates, with the outcomes only released back to the investigators following the completion of all testing. Following completion of the second round of IMTP testing, the equation developed in Part A was applied to provide a predicted 3RM squat value for each subject. Predicted and actual 3RM values were then compared using a paired two-tailed *t*-test to test the equation’s accuracy.

#### 2.3.2. Anthropometry

All anthropometric measures were recorded as per International Society for the Advancement of Kinanthropometry (ISAK) standards and carried out by individuals with a minimum ISAK Level One qualification. Anthropometric assessments were taken prior to the strength tests if they occurred on the same day. Measures taken included height, weight, seated height, leg length (calculated by subtracting seated height from height), arm span, and biacromial breadth. These measures were selected to account (indirectly) for the anthropometric variation in the subjects, which may influence force production during the IMTP and 3RM squats, including positioning, lever lengths, and grip width [22].

#### 2.3.3. Isometric Mid-Thigh Pull

The isometric mid-thigh pull (IMTP) was completed using a specialised power rack with adjustable pins to achieve the correct height positioning for each athlete. The bar was secured in place at the correct height in a mid-thigh position with the athlete’s knee and hip angles within 125°–145° and 155°–165°, respectively, as described previously [17,23]. The IMTP was performed using dual force plates (0.60 × 0.40 m; Model 10 kN 9286 B, Kistler Instrument Corporation, Amherst, NY, USA) sampling at 1000 Hz and analysed using Templo software (Version 2016.1.404 Contemplas GmbH, Kempten, Germany). Following the warm-up, athletes were instructed to complete two submaximal pulls at approximately 70–80% of their maximum. During submaximal efforts, corrections to technique and positioning were offered to the athletes where required, as part of the familiarisation process, with athletes asked to report any pain or discomfort during the submaximal efforts. Following a 120 s rest period, after submaximal efforts, athletes then completed two maximum effort pulls with a 120 s rest interval between each pull. Subsequent trials were conducted only if the two trials completed had a variation in peak force of greater than 200 N [17]. All IMTP trials were conducted by the same researcher using the standardised instructions “pull as hard and fast as possible by pushing the ground away” with a countdown “3, 2, 1 Pull” to increase the reliability of results. All strength tests and submaximal efforts were completed barefoot to reduce the influence of variation in shoe type [24], however, lifting straps were not used at any time. Onset of contraction was determined following the recommendations of Dos’Santos et al. [25]. Peak force and time-dependent force at 50, 100, 150, 200, and 250 ms were selected to inform the equation based on previous research using the IMTP establishing a strong relationship between these variables and similar strength movements [15].

#### 2.3.4. 3RM Squat

The 3RM squat test was completed by coaches with a minimum Level Two Strength and Conditioning Coach qualification accredited by the Australian Strength and Conditioning Association. Coaches determined their own warm-up protocol for athletes or used the warm-up protocol outlined in the above section on the IMTP; however, they were required to include at least two submaximal efforts to identify any potential risk of injury before performing the 3RM test. During these submaximal efforts, verbal feedback for technique and corrections were allowed. All 3RM squat testing was performed using a standard Olympic bar and weights in a squat rack. Athletes removed the bar from the rack in an upright position and moved backwards, ready to commence the back squat. Efforts were deemed valid if the athlete lowered the body to a depth where the thighs were parallel to the floor, returned to the standing position, and athletes completed three continuous repetitions. Athletes were then assisted to place the weight back onto the rack if required. Each athlete was given a minimum 120 s rest interval between trials. 

### 2.4. Statistical Analysis

#### 2.4.1. Part A

The data were inspected for assumptions of independence of error, linearity, multicollinearity, and outliers, while the dependent variable (3RM squat) was transformed using the square root method to meet the assumption of homoscedasticity. Variables used in the initial analysis included peak force, time-dependent force at 50, 100, 150, 200, and 250 ms, weight, standing height, seated height, leg length, biacromial breadth, and arm span. To establish the relationship between the predictor variables and the 3RM squat, Pearson’s correlation coefficient and multiple forward stepwise regression analysis were used, with correlations defined as strong (≥0.50–1), moderate (0.3–0.49), and low (≤0.29). The stepwise regression criteria for the inclusion of variables were an F to enter ≤0.050 and an F to remove ≥0.100. The accuracy of the equation was determined using a correlation coefficient with an adjusted r^2^ value and the standard error of the estimate (SEE) between the measured and predicted 3RM squat. 

#### 2.4.2. Part B

Following the development of the regression model, the equation was applied to the data obtained following training to predict each athlete’s 3RM squat value. The data were inspected for assumptions of normality and outliers prior to analysis, with no problems identified. Similarities between the predicted values of the equation established in Part A and an actual 3RM squat were assessed using a Pearson’s correlation and paired *t*-test. 

All data were analysed using the SPSS statistical analysis software program (IBM, Version 24.0), with statistical significance accepted as an alpha level of *p* ≤ 0.05. The statistical power using an *n* = 34 sample and an α *p* ≤ 0.05 results in a β of 0.8469.

## 3. Results

The absolute values for all performance outcome and predictor variables are provided in Table 2.

### 3.1. Part A

Prior to generating the regression equation, several assumptions were required to be met. A linear relationship between the dependent and independent variables was confirmed by partial regression plots and a plot of studentized residuals against predicted values. Analysis indicated an independence of residuals, as assessed by a Durbin–Watson statistic of 2.264. While homoscedasticity, as assessed by visual inspection of a plot of studentized residuals versus unstandardised predicted values, was only achieved following transformation of the dependent variable (during analysis referred to as 3RMsquat_sqrt). Multicollinearity was not apparent, as assessed by tolerance values > 0.01, while the assumption of a normal distribution of the data were met, as assessed by a Q-Q plot. 

The Pearson correlation between predictor and raw dependent variable ranged between r = 0.045 to r = 0.386 and r = −0.034 to r = −0.41 for positive and negative correlations, respectively. In comparison, after correction for homoscedasticity, this Pearson correlation range was r = 0.037 to r = 0.383 and r = −0.038 to r = −0.32 (Table 3). 

The linear stepwise regression analysis identified the predictors, which resulted in the maximum explanation of the variation in the dependent variable 3RMsquat_sqrt, as peak force and force at 100 ms (Table 4). Peak force and force at 100 ms statistically significantly predicted 3RMsquat_sqrt values, F(2,31) = 5.389, *p* = 0.010, adj. r^2^ = 0.210. The overall model r^2^ = 25.8% with an adjusted r^2^ of 21%. The regression coefficients and standard errors are presented in Table 5.

Using the stepwise regression analysis and identified coefficients and factors of peak force and force at 100 ms, the following equation was determined to predict 3RM squat:Predicted 3RM squat (kg) = [6.102 + (Peak Force × 0.002) − (Force at 100 ms × 0.001)]^2^(1)

### 3.2. Part B

The difference scores between predicted and actual 3RM squat values were normally distributed as confirmed by a Shapiro–Wilk test (*p* = 0.823) and no outliers were identified as assessed by the inspection of a boxplot. The Pearson’s correlation between predicted and actual 3RM squat values was *r* = −0.086, *p*= 0.771 (Figure 1A). The mean ± standard deviation of the predicted and actual 3RM squat values were 75.34 ± 12.15 and 71.38 ± 8.25, respectively. The mean difference was 3.95 (95% CI −4.18–12.09). There was no statistically significant difference between predicted and actual values, *t*(13) = 1.050, *p* = 0.313. Absolute differences between predicted and actual results are displayed in Figure 1B.

## 4. Discussion

Considering the growing participation of females in sport at the elite level, specific research incorporating methods of measuring and predicting strength to inform training load and exercise prescription is warranted in the female athlete population. Statistical outcomes from this investigation suggest that a regression equation using peak force and force at 100 ms achieved during the IMTP is statistically capable of predicting the 3RM squat in a female athlete cohort. However, differences between predicted and actual values were greater than 10 kg in a number of individual cases (Figure 1B). The observed difference between predicted and actual values poses a question on the practical suitability versus statistical suitability of the equation’s application. Interestingly, Materko et al. [26] previously reported in a mixed-sex cohort that the predicted and actual 1RM values in the leg curl and leg abduction based on fat-free mass (assessed from the sum of seven skinfolds) were not statistically significantly different. However, these authors also reported a standard error of 15.5%, which, in an athletic setting, translates to an error in excess of 10 kg for a 70 kg lift and could pose significant consequences. When applying the equation reported in this investigation, it is likely that 10 kg is an unacceptable level of error to replace the 3RM back squat considering the typical strength gain over the 6–8-week training period was 3–5 kg. The observed error may be reflective of the data transformation process used to determine the equation coefficients in our force output variables, as small errors between predicted and actual values are magnified during the back transformation process. Interestingly, the magnitude of our reported error is similar in magnitude to the error reported by Caven et al. [27] (9.7 kg) when applying a minimum velocity threshold loss approach with a two-point method to resistance-trained female athletes. While our results extend previous research supporting the relationship between dynamic and isometric force production, our results are not consistent with findings from other investigators using the IMTP in male subjects [15,28]. Thus, the work reported again highlights the need for more systematic evidence in female athletic cohorts to support the knowledge and programming requirements of these athletes [29].

In this cohort of sub-elite and elite-development female athletes, we did not observe a similar strength of relationship between peak force and absolute 3RM squat strength (Table 2) as reported by Haff et al. [17] using female weightlifters in the snatch (r = 0.93, *p* < 0.01) and clean-and-jerk (r = 0.64, *p* < 0.05). While there is a paucity of research investigating female athletes [29], in recreational male athletes, McGuigan et al. [15] reported a significant correlation (r = 0.97, *p* < 0.05) between IMTP and 1RM squat. Similarly, De Witt et al. [30] reported the deadlift 1RM (r = 0.88, *p*< 0.01) correlated strongly with the IMTP. Furthermore, in dynamic movement tasks peak IMTP force has also been observed to significantly correlated with concentric power (r = 0.52, *p* < 0.01) in the counter movement jump while force at 100 ms was highly correlated with counter movement jump height (r = 0.43, *p* < 0.01) and 10 m sprint time (r = −0.68, *p* < 0.01) in male rugby players [28]. The lack of a strong relationship between peak force and 3RM squat in this investigation could be related to the IMTP joint positioning, as identified by Beckham [31], who reported translation was more accurate when joint angles in the IMTP are similar to that used in dynamic movement. Peak force in the back squat has been reported at knee and hip angles of 104–115° and 119–125°, respectively [32], which are markedly lower in comparison to the joint angles performed at in the IMTP in this study of 125°–145° and 155°–165°, respectively. A stronger relationship may be observed in the future if the IMTP is completed at these lower hip and knee angles, where peak joint moment occurs in the dynamic movement [32], with potentially more translation from the IMTP force production to the back squat at these angles. The joint positions used in this study were informed by previous research eliciting the greatest peak force values in the IMTP [17,23] and may have been unfamiliar to the athletes involved, even though they all had suitable squat and deadlift experience. This supposition supports Comfort et al. [33], who proposed that due to the statistical similarities and high reliability found in multiple joint angles (knee at 120°, 130°, 140°, and 150°, and hip at 125° and 145°), performance of the IMTP should occur in the athlete’s preferred joint position. Whilst the intention of this investigation was to validate the use of the IMTP as a tool to monitor training and strength changes in addition to the predictive equation, perhaps the use of a 1RM back squat would have produced a stronger correlation and thus predictive capacity [34]. A 1RM back squat, however, still carries the same implications for time efficiency, residual soreness, and the potential for injury.

A further confounding factor to our poor predictive outcome is our inability to dissociate the resistance training experience of athletes from the outcome, as it should be expected that more experienced athletes would have a well-established technique and thus be able to perform the movements with a more ‘true’ measure of maximal strength [35]. Athletes included in this investigation were young (17.5 ± 2.5 years) with less than 2 years’ experience in heavy resistance training, making it plausible that although subjects were familiar with the test, maximum strength as measured in the 3RM squat may have been hindered by the technical requirements of the dynamic movement in contrast to the isometric strength performance required in the IMTP. This may explain the near-perfect correlations seen by Haff et al. [17] in international-level female Olympic weightlifters with the highest levels of technical proficiency and lifting experience. However, we advocate for the approach to be used by practitioners working with similar female team sport cohorts due to the ease of and reduced time commitment required for testing and because our collective sample includes various female team sport athletes.

The stepwise regression using peak force and force at 100 ms was only able to explain 26% of the variance, suggesting there may have been a missing component. This investigation included the collection of anthropometric characteristics; however, they were not included in the predictive equation by the stepwise regression analysis. Due to relevant ages, training ages, and the technique demanded by the back squat movement, greater strength may be achieved through a lifting technique that compensates for the differences in anatomical structure in this study. This is in contrast with the other literature describing the effects of anthropometry on strength and predicting RM strength using anthropometry; however, these were all conducted with male athletes [22,26,36]. Based on the previous literature in strength prediction, we assumed that the use of anthropometry and in particular body composition values, such as the sum of seven skinfolds, was likely to provide improved accuracy when used in addition to other methods. Whilst the sum of seven skinfolds has been linked to strength as a representation of muscle mass, it was not included in this investigation due to the subject demographic of young females that displayed body dysmorphia issues. This could potentially have affected the predictive accuracy of the equation, and its inclusion should be considered in future research or as an alternative body composition measure, such as factors derived from dual X-ray absorptiometry (DXA). There was a negative, moderate correlation between biacromial breadth and 3RM squat values (R = −0.41), suggesting that athletes with narrow shoulders lifted greater amounts, which is counter-intuitive. A ratio of biacromial breadth and height may be a more informative representation of athlete somatotype related to strength in female athletes, as it is expected that tall, broad athletes will have large, strong levers capable of producing more force.

The 6–8-week training period was identified as sufficient for a training adaptation to occur [21] and to elicit a change in 3RM squat and IMTP values to then validate the predictive equation. The training programs were not controlled by the investigators but rather by each team’s respective strength and conditioning coach due to the nature of recruiting athletes from national development teams, and therefore the prescription in this period could have elicited an increase in peak force (an indicator of strength) or force at 100 ms (an indicator of power) independent of one another, as a result of that athlete’s training focus. Importantly though, we did seek to control of the diurnal rhythm for strength performance [37], with the strength tests conducted between the hours of 2 pm and 6 pm. In future studies investigators, where possible, in real-world applied training settings should seek to at the very least enable a uniform training focus on strength or power development. Future research should be considered to investigate the applicability of the IMTP for RM prediction in other sports such as track and field events and female athletes that are of a more physiologically mature age with greater lifting experience and technical proficiency. As far as the authors are aware, this is the first study of its kind using the IMTP to predict lower-body strength in females. However, it is beyond the scope of this study to predict how the use of multiple regression analysis will relate to other populations of athletes from different sports. Performing the IMTP produces an output with a wide range of force–time characteristics that can provide insight for coaches into force development, rate of force development, and left and right imbalances, addressing components of strength and power that may inform prescription and would otherwise go unnoticed in a standard RM test. As the current study has not shown a relationship between results and an absolute load (kg) value for the back squat, the equation derived in this study does not appear suitable for use to predict the 3RM back squat within an acceptable range.

## 5. Practical Application

This investigation is the first to develop and then confirm the suitability of a statistically acceptable strength prediction equation in an all-female athletic team sport cohort. Although the 3RM squat can be statistically predicted using the variables of peak force and force at 100 ms derived from the IMTP (Equation (1)) performance, the use of this equation can produce predicted values that differ in excess of 10 kg from actual values. The magnitude of this difference is generally an unacceptable margin given the typical increase of 3–5 kg within the 6–8-week training period seen in this cohort. Therefore, it is recommended that the IMTP be used practically as an assessment of the athlete or personal training client independent of the 3RM back squat. Furthermore, if practitioners apply our reported equation to inform load prescription, particularly when working with young, developmental athletes in team sports, such as those represented in this study, it should be performed with the support of a secondary prescription method, such as repetitions in reserve (RiR), to help avoid instances of under- or overload prescription.

## Figures and Tables

**Figure 1 sports-12-00230-f001:**
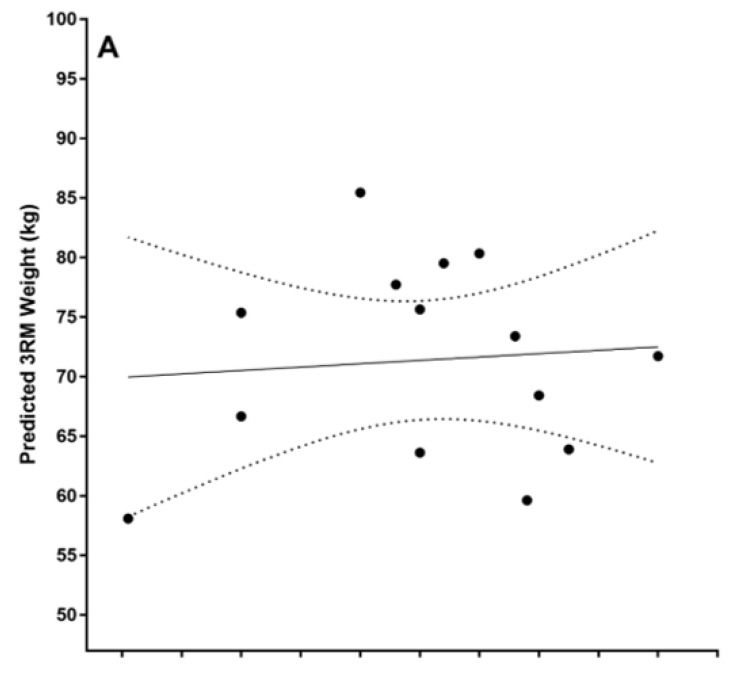
Scatter dot plot of predicted versus actual 3RM squat values, (**A**) with a correlation line of best fit (solid line) and 95% confidence interval (dotted line). Bland–Altman plot of the differences between predicted and actual values (**B**).

**Table 1 sports-12-00230-t001:** The anthropometric characteristics of athletes [mean ± SD (range)].

Cohort	Age (years) Mean ± SD (Range)	Height (cm) Mean ± SD (Range)	Weight (kg) Mean ± SD (Range)	Seated Height (cm) Mean ± SD (Range)	Arm Span (cm) Mean ± SD (Range)	Biacromial Breadth (cm) Mean ± SD (Range)
Athletics (*n* = 1)	19	173.4	59.40	92.1	176.0	37.9
* Soccer (*n* = 8)	16.8 ± 1.8 (16–21)	165.3 ± 5.7 (155.6–174.9)	61.6 ± 8.9 (51.2–76.8)	88.0 ± 3.1 (84.3–93.9)	163.6 ± 9.1 (149.0–172.0)	35.9 ± 1.8 (33.4–38.5)
Volleyball (*n* = 5)	21.2 ± 4.6 (18–28)	179.2 ± 2.2 (177.5–182.1)	74.1 ± 5.5 (68.9–81.5)	95.9 ± 3.3 (92.5–100.2)	181.4 ± 6.8 (176.0–193.0)	40.9 ± 1.4 (40.1–43.4)
Basketball (*n* = 13)	16.6 ± 1.0 (15–18)	181.4 ± 9.1 (162.4–192.9)	71.4 ± 7.8 (57.8–82.6)	94.1 ± 4.0 (86.8–99.8)	181.7 ± 10.6 (163.0–204.0)	38.4 ± 1.4 (35.8–41.3)
* Netball (*n* = 7)	17.0 ± 0.8 (16–18)	178.1 ± 5.1 (169.4–183.7)	69.6 ± 12.6 (55.1–90.6)	93.0 ± 2.1 (91.0–96.3)	177.4 (169.0–184.0)	37.4 ± 1.1 (36.3–39.4)
All (*n* = 34)	17.5 ± 2.5 (15–28)	176.4 ± 9.2 (155.6–192.9)	68.8 ± 9.7 (51.2–90.6)	92.7 ± 4.2 (84.3–100.2)	176.3 ± 11.2 (149.0–204.0)	37.9 ± 2.1 (33.4–43.4)

* denotes sports in which athletes participated in part B.

**Table 2 sports-12-00230-t002:** Absolute (mean ±SD) and range for performance outcome and predictor variables.

Cohort	3RM (kg)	Peak Force (N)	Force at 50 ms (N)	Force at 100 ms (N)	Force at 150 ms (N)	Force at 200 ms (N)	Force at 250 ms (N)
All (*n* = 34)	75.5 ± 12 (51–95)	1977 ± 220 (1616–2330)	828 ± 207 (628–1333)	1039 ± 285 (716–1622)	1257 ± 253 (1088–1767)	1479 ± 248 (1207–1900)	1641 ± 217 (1343–1980)

**Table 3 sports-12-00230-t003:** Pearson’s correlations (R) between predictor variables and the raw dependent 3RM squat and transformed dependent 3RMsquat_sqrt.

Predictors	3RM Squat	3RMsquat_sqrt
Peak force (N)	0.386	0.383
Force at 50 ms (N)	−0.040	−0.038
Force at 100 ms (N)	−0.128	−0.132
Force at 150 ms (N)	−0.063	−0.068
Force at 200 ms (N)	−0.034	−0.041
Force at 250 ms (N)	0.045	0.037
Height (cm)	0.294	0.304
Weight (kg)	0.284	0.291
Seated height (cm)	0.260	0.271
Leg length (cm)	0.271	0.276
Arm span (cm)	0.242	0.243
Biacromial breadth (cm)	−0.41	−0.32

**Table 4 sports-12-00230-t004:** Linear stepwise regression output with Pearson’s correlation (R), coefficient of determination (r^2^), goodness of fit (adjusted r^2^), and standard error of the estimate (SEE).

Predictor Combination	*R*	r^2^	Adjusted r^2^	SEE
Peak force (N) Force at 100 ms (N)	0.508	0.258	0.210	0.706

**Table 5 sports-12-00230-t005:** Stepwise multiple regression analysis summary with unstandardised regression coefficient (B), standard error of the coefficient (SE_β_), and standardised coefficient (β).

	*B*	SE_β_	*β*
Constant	6.102		
Peak force (N)	0.002	0.001	0.546
Force at 100 ms (N)	−0.001	0.000	−0.372

## Data Availability

The raw data supporting the conclusions of this article will be made available by the authors on request to the corresponding author.

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
