# Peer review of "Using the Isometric Mid-Thigh Pull to Predict Three-Repetition Maximum Squat Values in Female Athletes"

_sports, 2024, doi:10.3390/sports12090230_

Round 1

Reviewer 1 Report

Comments and Suggestions for Authors

Dear authors,

I have had the opportunity to read your paper which aim to develop a prediction equation for the 3RM in female athletes using the isometric test and mid-thigh extraction. I found it interesting because advancing science from such an applicable perspective is valuable for both physical trainers and experts in general. The study is well-conducted and focuses on a sample composed solely of women, which adds value to the literature that has predominantly focused on male samples. Recent articles suggest that training processes for female populations should differ from those for males. In summary, my opinion of the article is very positive, and it is evident that a great deal of work has gone into it. Congratulations to the team of authors. However, as you will see below, there are certain points I would like to highlight for the authors to consider. These are minor suggestions, and I believe a quick read will help understand my ideas and, if the authors deem it appropriate, incorporate them into the text.

Abstract

Regarding the abstract, I suggest breaking down the number of participants by sport, not just the total. Additionally, the abstract should explain the necessity and utility of the prediction equation. While experts might understand its usefulness, the general reader might not. It is mentioned that the equation is for controlling athlete fatigue, injury risk, and training load, but the leap to the prediction equation is unclear. What exactly is being predicted?

Introduction

In the introduction, the first paragraph mentions the difference between male and female sports and training, which is interesting but should be placed later in my opinion. The second paragraph is well-written and should come first, as it provides a general overview before delving into specifics about male and female sports. This reordering would make the text more natural and easier to read. The third paragraph mentions studies correlating with sprinting and weightlifting for predicting training load traditionally. It would be helpful to specify whether these studies used male or female samples to close the context. What I am about to comment on can be applied here and in the discussion, which I have mentioned to some extent. In my opinion, a strong point of this article is that the sample consists entirely of women, and there should be some comparisons with other studies that reflect the extensive literature on men or mixed samples, where some results have been generalized. Currently, it is observed that training methods between men and women need to be different. If the authors manage to convey this idea in the manuscript, perhaps with the reorganization I am suggesting, the value of the manuscript could increase.

Methodology

The first point in the methodology discusses the experimental approach to the problem, which is appreciated but should be included in the design section, which is currently missing from the manuscript. The methodology should follow a logical order: participants, design, and then procedure. The information about participants is comprehensive and helps understand the sample configuration. However, the section on RM Squat should better defend the claim that the sample did not need familiarization because they had experience. This information should be included in the participants section, specifying their experience with the exercise, whether they are semi-professional or professional, or have many years of sports practice, some data that indicate that the participants are familiarized with this exercise.

Results

The results section, along with the statistical analysis, is well-written. The tables are appropriate, but tables 3 and 4 might not be necessary and could be left to the authors’ discretion for aesthetic reasons. Overall, the results are clearly expressed and understandable.

Discussion

The discussion section is extensive and well-executed. Congratulations to the authors for their excellent work in this part. However, I believe there is room for improvement by comparing the results with other studies. The discussion is generally correct and does not require significant changes.

As mentioned in the introduction, it is essential to highlight the value of the sample, which consists solely of women. This article’s uniqueness lies in its female-only sample. Therefore, part of the discussion should focus on comparing this work with studies involving male or mixed samples. This comparison would underscore the article’s applicability to fitness training and physical preparation for sports.

The discussion should also emphasize the practical application of the training method and calculation in the female sample. The sample includes various sports with a common element of collective sports, which are similar in nature. This aspect is valuable for physical trainers working with female athletes across different sports. Including comparisons with previous studies on male or mixed populations would enhance the discussion’s relevance and applicability.

In the practical application section, it is crucial to address why this prediction equation is valuable and why physical trainers should consider using it over traditional methods. The conclusion mentions that this is the first study using this protocol with a female sample. Highlighting this point would add significant value to the article, helping the target audience understand why they should adopt this new approach.

In conclusion, the discussion should integrate comparisons with previous studies and emphasize the practical application of the findings. This approach would enhance the article’s value and relevance to physical trainers and experts in the field. Congratulations to the authors on their excellent work, and I hope these suggestions help further improve the manuscript.

Author Response

Comment 1: I have had the opportunity to read your paper which aim to develop a prediction equation for the 3RM in female athletes using the isometric test and mid-thigh extraction. I found it interesting because advancing science from such an applicable perspective is valuable for both physical trainers and experts in general. The study is well-conducted and focuses on a sample composed solely of women, which adds value to the literature that has predominantly focused on male samples. Recent articles suggest that training processes for female populations should differ from those for males. In summary, my opinion of the article is very positive, and it is evident that a great deal of work has gone into it. Congratulations to the team of authors. However, as you will see below, there are certain points I would like to highlight for the authors to consider. These are minor suggestions, and I believe a quick read will help understand my ideas and, if the authors deem it appropriate, incorporate them into the text.

Response: The authors thank the reviewer for their time providing insightful and helpful comments on our manuscript. Please see below for our point by point responses.

Comment 2: Regarding the abstract, I suggest breaking down the number of participants by sport, not just the total. Additionally, the abstract should explain the necessity and utility of the prediction equation. While experts might understand its usefulness, the general reader might not. It is mentioned that the equation is for controlling athlete fatigue, injury risk, and training load, but the leap to the prediction equation is unclear. What exactly is being predicted?

Response: We appreciate the reviewer’s suggestion regarding including the breakdown of the number of participants per sport in the abstract rather than just the total, however we are conscious of the abstract length and that including this information in the abstract does not further aid in interpreting the study outcomes. Thus, we decided not to make this change.

We have sought to clarify the reasons for the prediction equation and define more precisely what is actually being predicted. However, to clarify the statement made in the abstract concerning minimal fatigue inducement, this statement is not related to the prediction equation directly but does relate to the tests used by coaches and practitioners in setting, modifying and monitoring training prescription. Thus, if disruption and fatigue can be lessened by reducing the number of tests performed by the athlete, then this would be a positive.  Please see L14 and 27-29

Comment 3: In the introduction, the first paragraph mentions the difference between male and female sports and training, which is interesting but should be placed later in my opinion. The second paragraph is well-written and should come first, as it provides a general overview before delving into specifics about male and female sports. This reordering would make the text more natural and easier to read. The third paragraph mentions studies correlating with sprinting and weightlifting for predicting training load traditionally. It would be helpful to specify whether these studies used male or female samples to close the context. What I am about to comment on can be applied here and in the discussion, which I have mentioned to some extent. In my opinion, a strong point of this article is that the sample consists entirely of women, and there should be some comparisons with other studies that reflect the extensive literature on men or mixed samples, where some results have been generalized. Currently, it is observed that training methods between men and women need to be different. If the authors manage to convey this idea in the manuscript, perhaps with the reorganization I am suggesting, the value of the manuscript could increase.

Response: We thank the reviewer for their valuable commentary to improve the flow and importance of our work to the readership. We have restructured the paragraph order as suggested and indicated the sex of cohorts referred to (see L145).

Comment 4: The first point in the methodology discusses the experimental approach to the problem, which is appreciated but should be included in the design section, which is currently missing from the manuscript. The methodology should follow a logical order: participants, design, and then procedure. The information about participants is comprehensive and helps understand the sample configuration. However, the section on RM Squat should better defend the claim that the sample did not need familiarization because they had experience. This information should be included in the participants section, specifying their experience with the exercise, whether they are semi-professional or professional, or have many years of sports practice, some data that indicate that the participants are familiarized with this exercise.

Response: While we appreciate the reviewers comment on the order of the methodology section, this is not a journal requirement, and it is the authors preference to keep the order as submitted. The Experimental Approach to the Problem section is describing the experimental design which in our opinion then provides context to the participants recruited for the investigation.

We thank the reviewer for the suggestion to better justify the cohort’s experience and familiarisation with the primary strength assessment. We have expanded on our explanation and moved the information as suggested, please see L231-235.

Comment 5: The results section, along with the statistical analysis, is well-written. The tables are appropriate, but tables 3 and 4 might not be necessary and could be left to the authors’ discretion for aesthetic reasons. Overall, the results are clearly expressed and understandable.

Response: Thank you, we understand the reviewer’s opinion regarding the inclusion of Tables 3 and 4, considering that there was no negative commentary from Reviewers 2 and 3, the authors have decided to retain the Tables.

Comment 6: The discussion section is extensive and well-executed. Congratulations to the authors for their excellent work in this part. However, I believe there is room for improvement by comparing the results with other studies. The discussion is generally correct and does not require significant changes.

Response: Thank you, we are glad that you enjoyed reading the work

Comment 7: As mentioned in the introduction, it is essential to highlight the value of the sample, which consists solely of women. This article’s uniqueness lies in its female-only sample. Therefore, part of the discussion should focus on comparing this work with studies involving male or mixed samples. This comparison would underscore the article’s applicability to fitness training and physical preparation for sports.

The discussion should also emphasize the practical application of the training method and calculation in the female sample. The sample includes various sports with a common element of collective sports, which are similar in nature. This aspect is valuable for physical trainers working with female athletes across different sports. Including comparisons with previous studies on male or mixed populations would enhance the discussion’s relevance and applicability.

Response: Thank you for the suggestion, we have sought to where appropriate strengthen the reference to our unique athlete cohort (L436-443 and L543-546).

Comment 8: In the practical application section, it is crucial to address why this prediction equation is valuable and why physical trainers should consider using it over traditional methods. The conclusion mentions that this is the first study using this protocol with a female sample. Highlighting this point would add significant value to the article, helping the target audience understand why they should adopt this new approach.

Response: Thank you for the suggestion as detailed in the previous response we have strengthened our language describing why it is valuable for practitioners to use the reported prediction equation (L543-546). We have also highlighted the female sample in the practical application section (L615-627)

Comment 8: In conclusion, the discussion should integrate comparisons with previous studies and emphasize the practical application of the findings. This approach would enhance the article’s value and relevance to physical trainers and experts in the field. Congratulations to the authors on their excellent work, and I hope these suggestions help further improve the manuscript.

Response: Thank you for your suggestions they have helped us to strengthen the relevance of our outcomes to the readership

Reviewer 2 Report

Comments and Suggestions for Authors

Dear Authors,

I have reviewed your manuscript, "Using the Isometric Mid-Thigh Pull to Predict 3RM Squat Val-2 ues in Female Athletes." This study aimed to develop a prediction equation in female athletes for the three-repetition maximum (3RM) squat using the isometric mid-thigh pull and basic anthropometric assessments.

I want to congratulate you on the research you have conducted. It is a very interesting study. After completing the revision of the manuscript, I would like to make a few comments:

1)       Abstract: The abstract is clear and well-presented.

2)       Introduction: It was well explained.

3)       Methods Section: The methods were well explained, and the statistical analysis applied was described appropriately and aligned with the study's objectives.

However, there seems to be a gap in the sample as they are different sports and have different strength development and training forms. This may have led to the incongruous results presented in the study, which the authors mentioned during the Discussion.

On the other hand, strength development and training may have influenced these results!

4)       Discussion: It is well presented.

5)       Conclusions: Agree with the results.

Despite incongruous results, once again, I congratulate you on your work. I hope you will consider my comments.

Best regards

Author Response

Comment: I have reviewed your manuscript, "Using the Isometric Mid-Thigh Pull to Predict 3RM Squat Values in Female Athletes." This study aimed to develop a prediction equation in female athletes for the three-repetition maximum (3RM) squat using the isometric mid-thigh pull and basic anthropometric assessments.

I want to congratulate you on the research you have conducted. It is a very interesting study. After completing the revision of the manuscript, I would like to make a few comments:

1)       Abstract: The abstract is clear and well-presented.

2)       Introduction: It was well explained.

3)       Methods Section: The methods were well explained, and the statistical analysis applied was described appropriately and aligned with the study's objectives.

However, there seems to be a gap in the sample as they are different sports and have different strength development and training forms. This may have led to the incongruous results presented in the study, which the authors mentioned during the Discussion.

On the other hand, strength development and training may have influenced these results!

4)       Discussion: It is well presented.

5)       Conclusions: Agree with the results.

Despite incongruous results, once again, I congratulate you on your work. I hope you will consider my comments.

Response: Thank you for the feedback and your time in reviewing our manuscript. We have sought to incorporate your comment from the methods section into our discussion and the limitations of the interpretation.

Reviewer 3 Report

Comments and Suggestions for Authors

General comments

Thank you for the opportunity to review this interesting manuscript, I congratulate the authors on a solid manuscript that does require some minor amendments and additions to the text to make the outcomes more impactful. I think the clarity on the ineffectiveness of the prediction equation is useful as many authors who do this (such as with VBT) don’t commonly highlight these limitations of loading.

Specific comments

Abstract

L9-11 – You need to add some punctuation in here, it is a very long sentence.

L18 – Change “@” for “at”

L20-21 – Could you clarify if the r values here are Pearson’s correlation values or coefficient of determination, if the former please make the R lower case.

L27 – It would be useful if you could make the 10kg relative to the 3RM.

Introduction

This is a nice introduction I think some of the paragraphs need to use some more literature for exploration. I am not going to be too explicit on where, but I think they could be beneficial.

Paragraph 1 –

Hunter SK, S Angadi S, Bhargava A, Harper J, Hirschberg AL, D Levine B, L Moreau K, J Nokoff N, Stachenfeld NS, Bermon S. The Biological Basis of Sex Differences in Athletic Performance: Consensus Statement for the American College of Sports Medicine. Med Sci Sports Exerc. 2023 Dec 1;55(12):2328-2360. doi:

Comfort P, McMahon JJ, Lake JP, Ripley NJ, Triplett NT, Haff GG. Relative strength explains the differences in multi-joint rapid force production between sexes. PLoS One. 2024 Feb 15;19(2):e0296877. doi: 10.1371/journal.pone.0296877. PMID: 38359039; PMCID: PMC10868802.

Paragraph 2

L52 – Please change “Olympic lifts” to “weightlifting”, Olympic only refers too lifting performed in the Olympics

Paragraph 3

Tan WZN, Lum D. Predicting 1 Repetition Maximum Squat With Peak Force Obtained From Isometric Squat at Multiple Positions. J Strength Cond Res. 2024 Jul 23. doi: 10.1519/JSC.0000000000004849. Epub ahead of print. PMID: 39074205.

Methods

L104 – What do you mean by “sport specific”

L108-109 – Use the below reference to define populations.

McKay AKA, Stellingwerff T, Smith ES, Martin DT, Mujika I, Goosey-Tolfrey VL, Sheppard J, Burke LM. Defining Training and Performance Caliber: A Participant Classification Framework. Int J Sports Physiol Perform. 2022 Feb 1;17(2):317-331. doi: 10.1123/ijspp.2021-0451. Epub 2022 Dec 29. PMID: 34965513.

L160 – Were straps used for the IMTP?

L174- Why 200N as this contrasts what has been suggested in previous literature by Comfort et al? They suggest using 250N, 200 N is arguably a more robust method but some justification would be useful (maybe familiarization).

L179 – How was onset identified for the time related metrics?

Results

Please could the authors provide absolute values (mean and SD) for the data.

Discussion

L275-278 – I don’t suitable is the correct word here, especially as you then proceed to explain why it is not suitable. Maybe just suggest you can use the peak force and force at 100ms to predict 3RM, maybe statistical capability even if not practical suitability.

L275 please change “@” to “at”

L305-306 – Would an iso-squat be more appropriate? Potentially due to slightly altered knee and hip angles, but pain related inhibition from the bar on the traps with limited muscle to pad or the use of foam pad which would negate some of the time related metrics could be an issue.

L310 – You have missed an abbreviation for the isometric mid-thigh pull here.

L341 – Check the reference here, the presentation looks off.

L346 – Although interesting to consider body composition, for a dynamic task they would still have to move that mass.

L358-L363 – This is a huge limitation, not knowing what was performed does limit the applicability of part 2. Is there any data you could attain e.g. exercises and such. It just needs more detail and clarity.

Author Response

Comment 1: Thank you for the opportunity to review this interesting manuscript, I congratulate the authors on a solid manuscript that does require some minor amendments and additions to the text to make the outcomes more impactful. I think the clarity on the ineffectiveness of the prediction equation is useful as many authors who do this (such as with VBT) don’t commonly highlight these limitations of loading.

Response: The authors thank the reviewer for their time providing insightful and helpful comments on our manuscript. Please see below for our point by point responses

Comment 2: L9-11 – You need to add some punctuation in here, it is a very long sentence.

Response: Thank you we have separated the sentence into two sentences, please L10.

Comment 3: L18 – Change “@” for “at”

Response: Changed as requested

Comment 4: L20-21 – Could you clarify if the r values here are Pearson’s correlation values or coefficient of determination, if the former please make the R lower case.

Response: Yes, apologies these are Pearson’s correlation values and the R has been changed to lower case.

Comment 5: L27 – It would be useful if you could make the 10kg relative to the 3RM.

Response: We appreciate the reviewers suggestion but the 10kg value is relative to the standard error of the prediction of the 3RM, and the prior percentage value which is more accurate provides the reader with the ability to make a relative judgement calculation to values displayed in figure 1, y axis. No change has been made.

Comment 6: This is a nice introduction I think some of the paragraphs need to use some more literature for exploration. I am not going to be too explicit on where, but I think they could be beneficial.

Paragraph 1 –

Hunter SK, S Angadi S, Bhargava A, Harper J, Hirschberg AL, D Levine B, L Moreau K, J Nokoff N, Stachenfeld NS, Bermon S. The Biological Basis of Sex Differences in Athletic Performance: Consensus Statement for the American College of Sports Medicine. Med Sci Sports Exerc. 2023 Dec 1;55(12):2328-2360. doi:

Comfort P, McMahon JJ, Lake JP, Ripley NJ, Triplett NT, Haff GG. Relative strength explains the differences in multi-joint rapid force production between sexes. PLoS One. 2024 Feb 15;19(2):e0296877. doi: 10.1371/journal.pone.0296877. PMID: 38359039; PMCID: PMC10868802.

Response: Thank you for the suggestion, where suitable we have included this literature.

Comment 6: L52 – Please change “Olympic lifts” to “weightlifting”, Olympic only refers too lifting performed in the Olympics

Response: Change accepted

Comment 7: Paragraph 3

Tan WZN, Lum D. Predicting 1 Repetition Maximum Squat With Peak Force Obtained From Isometric Squat at Multiple Positions. J Strength Cond Res. 2024 Jul 23. doi: 10.1519/JSC.0000000000004849. Epub ahead of print. PMID: 39074205.

Response: We thank the review for highlighting this recent work and while we did not include in the introduction we have referred to the work in the discussion

Comment 8: L104 – What do you mean by “sport specific”

Response: To clarify for the reviewer, we are referring to a programmed training that was specific to sport, ie it was not equated and was individualised by the involved strength and conditioning coach. We have amended the wording to clarify this point, please see L224-225

Comment 9: L108-109 – Use the below reference to define populations.

McKay AKA, Stellingwerff T, Smith ES, Martin DT, Mujika I, Goosey-Tolfrey VL, Sheppard J, Burke LM. Defining Training and Performance Caliber: A Participant Classification Framework. Int J Sports Physiol Perform. 2022 Feb 1;17(2):317-331. doi: 10.1123/ijspp.2021-0451. Epub 2022 Dec 29. PMID: 34965513.

Response: This reference has been added

Comment 10: L160 – Were straps used for the IMTP?

Response: Thank you for our oversight in not clarifying, straps were not used by any participant.

Comment 11: L174- Why 200N as this contrasts what has been suggested in previous literature by Comfort et al? They suggest using 250N, 200 N is arguably a more robust method but some justification would be useful (maybe familiarization).

Response: We thank the reviewer for the comment however, in the original article by Haff etal they applied a 200N limit, we have now included the reference to support our justification.

Comment 12: L179 – How was onset identified for the time related metrics?

Response: Thank you for highlighting this omission from our methods, we have now described the procedure followed which was as recommended by Dos’Santos et al (L316-317)

Comment 13: Please could the authors provide absolute values (mean and SD) for the data.

Response: Absolute values have now been included as a new table, please see Table 2 L370

Comment 14: L275-278 – I don’t suitable is the correct word here, especially as you then proceed to explain why it is not suitable. Maybe just suggest you can use the peak force and force at 100ms to predict 3RM, maybe statistical capability even if not practical suitability.

Response: Thank you for the suggestion, we have amended the wording (L421).

Comment 15: L275 please change “@” to “at”

Response: Changed

Comment 16: L305-306 – Would an iso-squat be more appropriate? Potentially due to slightly altered knee and hip angles, but pain related inhibition from the bar on the traps with limited muscle to pad or the use of foam pad which would negate some of the time related metrics could be an issue.

Response: We thank the review for the comment however we preferred to refer to the potential to use of the isometric back squat later in the paragraph as we felt the discussion flowed more appropriately, and so we have made no further changes

Comment 17: L310 – You have missed an abbreviation for the isometric mid-thigh pull here.

Response: Changed

Comment 18: L341 – Check the reference here, the presentation looks off.

Response: Corrected

Comment 19: L346 – Although interesting to consider body composition, for a dynamic task they would still have to move that mass.

Response: We agree with the review that in a dynamic task the mass is still required to be moved, however the statement made relates to whether a more accurate and less invasive measure of body composition could have improved the predictive accuracy and capacity of the proposed equation. No changes have been made in response to this comment.

Comment 20: L358-L363 – This is a huge limitation, not knowing what was performed does limit the applicability of part 2. Is there any data you could attain e.g. exercises and such. It just needs more detail and clarity.

Response: Respectfully the authors disagree that this is a huge limitation to the application of the prediction equation. The training completed while possible to influence the time domain metrics, the training was not a variable included in the calculation. As we did not seek to control the training program completed which we have declared we do not have access to the data to include, as such no change is made.